# Topographic Pattern-Based Nomogram to Guide Keraring Implantation in Eyes with Mild to Moderate Keratoconus: Visual and Refractive Outcome

**DOI:** 10.3390/jcm14030870

**Published:** 2025-01-28

**Authors:** Ugo de Sanctis, Paolo Caselgrandi, Carlo Gennaro, Cecilia Tosi, Enrico Borrelli, Paola Marolo, Michele Reibaldi

**Affiliations:** 1Section of Ophthalmology, Department of Surgical Sciences, University of Turin, 10126 Turin, Italy; ugo.desanctis@unito.it (U.d.S.); caselgrandi.paolo@gmail.com (P.C.); carlo.gennaro@edu.unito.it (C.G.); borrelli.enrico@yahoo.com (E.B.); paola.marolo@unito.it (P.M.); 2Studio Oculistico de Sanctis, 10121 Turin, Italy; cecilia.tosi@ugodesanctis.com; 3Centro LasER, 10128 Turin, Italy

**Keywords:** cornea, corneal surgery, keraring, keratoconus, ICRS

## Abstract

**Background:** To assess the outcome of Keraring (Mediphacos, Brazil) implantation according to a topographic pattern-based nomogram in eyes with mild to moderate keratoconus. **Materials and Methods**: A topographic pattern-based nomogram was used to guide Keraring selection in 47 consecutive eyes with stage I-II keratoconus (Amsler-Krumeich staging), which underwent femtosecond laser-assisted implantation at a single center. Electronic data of LogMar uncorrected distance visual acuity (UDVA) and corrected distance visual acuity (CDVA) manifest refraction and tomographic analysis (Pentacam HR, Oculus, Germany) measured preoperatively and at the last postoperative examination were retrospectively analyzed. **Results:** Mean follow-up was 18.8 months. (range 3–35). Mean UDVA improved (*p* < 0.001) from 0.87 ± 0.27 to 0.35 ± 0.21. UDVA increased on average by 5.13 lines. Mean CDVA improved from 0.21 ± 0.10 to 0.09. ± 0.07, and the proportion of eyes with CDVA ≥ 20/25 increased from 29.8% to 85.1% after surgery. No eyes lost lines of CDVA. The Alpins correction index of astigmatism was 0.77 and the mean refractive cylinder decreased from 4.99 ± 1.89 to −2.31 ± 1.47 D (*p* < 0.001). Mean and maximal keratometry was reduced on average by −2.10 ± 1.42 D and −3.02 ± 3.68 D, respectively (*p* < 0.001). The RMS of corneal high-order aberrations dropped from 3.296 ± 1.180 µm to 2.192 ± 0.919 µm, and that of vertical coma from −2.656 ± 1.189 µm to −1.427 ± 1.024 µm (*p* < 0.001). All topometric indices improved after surgery. **Conclusions:** Planning Keraring implantation using the topographic pattern-based nomogram is very effective and safe in eyes with mild to moderate keratoconus. Using that nomogram of UDVA and CDVA are clinically significant.

## 1. Introduction

Intracorneal ring segments (ICRS) have long been used for the optical rehabilitation of eyes with keratoconus alone or in combination with other surgical procedures [1,2,3]. The implantation of ICRS at the midperiphery of the cornea generates an arc-shortening effect that reduces the steepening, asymmetry, and asphericity of the cornea, leading to improvement of the visual function [4,5]. The change of corneal curvature generated by the implant is related to different factors, which include depth of implantation and implant characteristics such as cross-sectional shape, thickness, arc length, and diameter.

There are, at present, four main variations of ICRS available internationally: Intacs and Intacs SB (Addition Technology, Inc., Lombard, IL, USA), Ferrara Ring (Ferrara Ophthalmics, Belo Horizonte, Brasil), Keraring (Mediphacos Ltd., Belo Horizonte, Brasil), and MyoRing (Dioptex GmbH, Linz, Austria). These devices are manufactured in different shapes and sizes. A widely used type of ICRS is the Keraring, which is made of polymethylmethacrylate (PMMA) and has a triangular cross-section [2,3]. It is manufactured in 67 variants of arc length thickness, and diameter. They also include models with asymmetrical thickness (Keraring AS), which have been tailored to treat keratoconic corneas with more pronounced asymmetry and have become available in recent years [6,7,8,9].

Different nomograms have been proposed over the years to plan Keraring implantation [10,11,12,13,14,15]. They provide clinical guidelines to determine the number of ICRS to implant, their arc lengths and thicknesses, as well as the location of insertion, based on preoperative parameters. Starting in 2018, the device manufacturer recommended using the nomogram developed by Fernandez-Vega and Alfonso that guides device selection according to the topographic pattern of keratoconus [13].

To the best of our knowledge, only 2 studies had reported the efficacy and safety of the procedure using that nomogram [6,9]. Those studies included keratoconic eyes with an asymmetrical topographic pattern. In this study we investigated the nomogram efficacy in a cohort of eyes with mild to moderate keratoconus where both asymmetrical and symmetrical topographic patterns were included to estimate the overall performance of the procedure in those disease stages.

## 2. Material and Methods

This single-center retrospective study included consecutive eyes that underwent Keraring implantation for keratoconus at the Las.E.R Refractive Surgery Clinic, Turin, Italy, in the period between 1 January 2019 and 30 June 2022. Only cases with keratoconus at stage I–II according to the Amsler-Krumeich classification, clear central cornea, contact lens intolerance, at least 3 months of follow-up and complete data available were included. Eyes with a history of ocular surgery, including corneal collagen cross-linking (CXL), that was performed before, in combination with, or after ICRS implantation were excluded. Eyes with amblyopia, herpetic disease, ocular media opacities, glaucoma, and retinal or neurophthalmic disease were also excluded. The study followed the tenets of the Declaration of Helsinki, and all patients were asked to sign an informed consent form before treatment.

### 2.1. Preoperative Protocol

All patients were examined before surgery at the clinic following a standardized protocol that included measurement of LogMar uncorrected distance visual acuity (UDVA) and corrected distance visual acuity (CDVA), manifest and cycloplegic refraction, slit-lamp examination, Scheimpflug corneal tomography (Pentacam HR, Oculus, Wetzlar, Germany), OCT corneal pachymetry (RTVue100, Optovue, Fremont, CA, USA), infrared pupillometry (Antares, CSO, Florence, Italy), and specular endothelial microscopy. Manifest refraction was performed following a standardized method by two examiners (U.d.S. and C.T.) who had previously undergone refraction training and validation with this method [16].

Preoperative planning of ICRS implantation was independently performed by the surgeon (U.d.S.) and by the Mediphacos Consulting following the guidelines of the Fernandez-Vega and Alfonso nomogram [12]. The Scheimpflug-derived sagittal map of the anterior corneal surface was analyzed to classify the topographic pattern in 5 categories (croissant, duck, snowman, nipple, bow-tie) according to the location of the cone, the relationship between axes (refractive, topographic, and coma), the orthogonality of astigmatism, and the symmetry of astigmatism. Then the number of the implants, their arc lengths, and thickness, as well as the location of insertion, were planned considering mean keratometry, steep keratometry, astigmatic error, and sphero-equivalent as reported in the nomogram. Differences in keratoconus pattern classification and surgical planning between surgeon and manufacturer consulting services were reviewed and further discussed.

### 2.2. Surgical Technique

The surgical procedure was carried out under sterile conditions and topical anesthesia by a single surgeon. The horizontal meridian (0–180°) of the cornea was marked at the slit lamp with the patient in an upright position. The intrastromal tunnel was created using the Visumax 500 femtosecond laser (Carl Zeiss Meditech, Jena, Germany). The tunnel depth was set at a depth of 75% of the corneal thinnest point in the hypothetical tunnel area as measured by means of the OCT pachymetry map. The inner and outer diameters of the tunnel were set at 4.8/6.2 mm for ICRS with a 5 mm optical zone and at 5.8/7.4 mm for those with a 6 mm optical zone. The implant was inserted inside the intrastromal tunnel using the Albertazzi forceps and the Sinskey hook. Tobramycin 0.3% and dexamethasone 0.1% eyedrops were used four times a day for 2 weeks, and preservative-free artificial tears were used four times a day for 1 month for postoperative treatment.

Follow-up visits were performed the day after surgery, 1 month, 3 months, 12 months after surgery, and every year thereafter. In the first postoperative day, measurement of UDVA and the slit-lamp evaluation of the cornea were performed. The other postoperative examinations also included measurement of CDVA, manifest refraction, and Scheimpflug corneal tomography.

### 2.3. Statistical Analysis

Statistical analysis was carried out using SPSS (Version 21.0; IBM Corp, Armonk, NY, USA).

Descriptive statistics were reported as mean and standard deviation for continuous variables and as frequency and percentage for qualitative variables. The normality of quantitative data was checked by the Shapiro-Wilk test. The paired samples *t*-test and its nonparametric equivalent (Wilcoxon signed-rank test) for variables with no Gaussian distribution were used to compare preoperative and postoperative variables. Categorical variables were evaluated using chi-square or, when necessary, Fisher’s exact tests. A *p*-value ≤ 0.05 was considered for statistical significance.

## 3. Results

Sixty-six eyes with stage I-II keratoconus received Keraring implantation in the period of study. From these, 19 eyes were excluded from analysis: 14 eyes, which underwent CXL before or simultaneously with ICRS implantation; 3 eyes with follow-up < 3 months; 1 eye with amblyopia; and 1 eye with cataract. Forty-seven eyes of 42 patients were included for analysis. The patients mean age was 35.4 ± 11.78 years (range 18–60). Thirty-seven patients received Keraring implantation in one eye, and 5 patients in both eyes. Keratoconus was stage I in 23 eyes and stage II in 24 eyes. The keratoconus pattern and the type of Keraring implanted are reported in Table 1. The mean follow-up was 18.8 ± 15.52, months and 26/47 eyes (55.3%) had a follow-up greater than 12 months.

### 3.1. Visual Outcome

UDVA increased (*p* < 0.001) from 0.87 ± 0.27 (range 1.20–0.40) to 0.35 ± 0.21 (range 1.00–0.00) after surgery. The cumulative percentage of preoperative CDVA versus postoperative UDVA is reported in Figure 1. The proportion of eyes with postoperative UDVA ≥ 20/40 and ≥ 20/25 was 52.2% and 15.2%, respectively. On average, UDVA increased by 5.13 ± 2.44 lines (range: 1–10). All eyes gained at least 1 line, and 55.3% of eyes gained five or more lines (Figure 2).

CDVA increased after surgery (*p* < 0.001) from 0.21 ± 0.0 (range 0.50–0.10) to 0.09 ± 0.07 (range 0.20–0.10). The cumulative percentage of preoperative CDVA versus postoperative CDVA is reported in Figure 3. All eyes had postoperative CDVA of at least 20/32. The proportion of eyes with postoperative CDVA ≥ 20/25 and ≥ 20/20 was 85.1% and 29.8%, respectively. No eyes lost lines of CDVA. The mean gain of CDVA afterward was 1.3 ± 0.75 lines (range: 0–3) (Figure 4).

### 3.2. Refractive Outcome

The preoperative and postoperative sphere, cylinder, and sphero-equivalent values are shown in Figure 5. The spherical error did not statistically change (*p* = 0.84) after surgery (+0.54 ± 2.43 D; range −5.25 to +5.25 preoperatively versus +0.63 ± 0.18 D; range −4.00 to +5.00 postoperatively). The refractive cylinder decreased (*p* < 0.001) from −4.99 ± 1.89 D (range −9.00 to −1.50) to −2.31 ± 1.47 (range −5.25 to 0.0), and the spherical equivalent (*p* = 0.001) from −1.93 ± 2.52 D (range −8.00 to +2.25) to −0.52 ± 1.81 (range −6.63 to +3.00).

The overview of Alpins vectorial analysis is shown in Figure 6. The correction index (ratio between surgically induced astigmatism and target-induced astigmatism) was 0.77.

### 3.3. Tomographic Data

The corneal tomographic data are reported in Table 2, Table 3 and Table 4.

The mean keratometric and asphericity values were significantly (*p* < 0.05) improved after surgery. The corneal thickness at the thinnest point was significantly (*p* = 0.016) increased (Table 2).

The mean values of corneal wavefront aberrations are shown in Table 3. The RMS of high-order aberrations and vertical coma were reduced (*p* < 0.001) after the intervention.

Mean values of the topometric indices are reported in Table 4. After surgery, ISV, IVA, IA, IHA, IHD, Rmin, and TKC were significantly improved.

The postoperative tomographic changes in 4 eyes with different preoperative topographic patterns are shown in Figure 7.

### 3.4. Complications

No serious complications were observed intraoperatively. After surgery, the ICRS was slightly repositioned inside the corneal tunnel to improve the refractive and tomographic results in two eyes. Infection, superficial erosion, or ICRS extrusion did not occur in any eye.

## 4. Discussion

Planning of ICRS implantation in eyes with keratoconus is guided by nomograms that have changed over the years to increase the predictability of results and improve the clinical outcome of the surgical procedure. In this study, the nomogram proposed by Fernandez-Vega and Alfonso was used to plan Keraring implantation in a cohort of patients with mild to moderate keratoconus [12,13]. This method is based on a classification of disease topographic pattern. It provides objective criteria to identify the topographical pattern. Following these criteria, the agreement between the examiner and the manufacturer consulting in classifying the disease was good (42/47 cases). In our cohort that included eyes with stage I-II keratoconus, the most common topographic pattern was the croissant type (59.6%). The duck-type and snowman-type patterns, which received ICRS with asymmetrical thickness, were found in 25.5% of eyes.

Planning Keraring implantation using the topographic pattern-based nomogram led to very good visual and refractive results. Although vector analysis showed a tendency to astigmatism undercorrection (correction index of 0.77), the cylindrical error was reduced by more than 50% on average (from −4.99 ± 1.89 D to 2.31 ± 1.47 D), and the proportion of eyes with cylindrical error ≥ 5 D decreased from 38.3% to 8.5% after surgery. The reduction of cylindrical error led to significant improvement of UDVA. It increased by more than 5 lines. on average, and postoperatively it was ≥20/40 in more than 50% of eyes. Moreover, the topometric and tomographic analysis highlighted a significant improvement of corneal regularity and asphericity towards more physiological values after surgery. High-order aberrations decreased significantly, and mean vertical coma dropped from −2.656 μm to −1.427 μm. The improvement of corneal symmetry and regularity allowed a more effective correction of the residual refractive error. The spectacle CDVA was ≥ 20/25 in 85% of eyes after surgery in comparison with 30% of eyes before surgery.

Femtosecond laser-assisted implantation of ICRS allows creation of the intrastromal tunnel at a very precise depth [17,18]. Using this technique, the risk of complications such as corneal perforation, superficial erosion, and ICRS extrusion is very low [19]. Moreover, in eyes with stage I-II keratoconus, the corneal thickness in the implantation area is usually above 500 microns, and the tunnel can be created at a depth that is within the pachymetric safety limits. In this study, the procedure was very safe. Using the Visumax 500 femtosecond laser, the tunnel was created with a suction time shorter than 10” and no suction loss occurred. No case of superficial erosion or extrusion was observed, and no eye had CDVA loss ≥ 1 line. The ICRS had to be repositioned a few weeks after surgery in 2 eyes because it was implanted not far enough from the incision during the first surgery.

Other nomograms have been proposed to guide Keraring implantation in eyes with keratoconus. Several studies reported the clinical outcome using the Keraring Calculation Guidelines 2009 [20,21,22]. It included three nomograms for keratoconic corneas with different distributions of the steep area. Using that method, the mean postoperative LogMar UDVA ranged from 0.50 to 0.54, and the mean LogMAR CDVA from 0.14 to 0.24 in eyes with stage I-III keratoconus [20,21,22]. Fariselli et al. developed an artificial neural network to guide Keraring implantation. After surgery, the mean LogMar UDVA and CDVA were 0.43 and 0.14, respectively, in 20 eyes with keratoconus [14]. In a prospective randomized controlled trial, Iqbal et al. used an asphericity based nomogram to plan implantation of Keraring (210° arc length) combined with transepithelial corneal cross-linking in 104 eyes with stage I-II keratoconus [15]. In the study group, postoperative LogMar UDVA and CDVA were significantly better than that achieved using the standard Keraring Calculation Guidelines 2009 (0.53 vs. 0.68 for UDVA and 0.18 vs. 0.21 for CDVA, respectively). The pattern-based nomogram performs at least as well as reported with other methods. In this study, UDVA was 0.35, which is >1 line compared to that reported with other methods, and CDVA was 0.09. However, differences in keratoconus severity, type of ICRS, and surgical technique between studies might have biased the results. Thus, prospective randomized comparative studies are required to assess if a topographic pattern-based nomogram may improve the clinical outcome of Keraring implantation in eyes with keratoconus as compared with other methods.

Several limitations of our study deserve discussion. The efficacy and safety of the nomogram were assessed in a small cohort of eyes. It did not include any case of central keratoconus with a nipple-type pattern that, on average, are more difficult to treat. Further studies including a larger cohort of patients will be needed to evaluate whether the nomogram efficacy is influenced by disease severity and topographic pattern. The accuracy of our estimates could also be reduced by the retrospective design of the study. However, rigorous eligibility criteria were applied, and a systematic protocol for data collection was adopted to reduce the risk of distortions.

In conclusion, Keraring implantation guided by a topographic pattern-based nomogram is very effective and safe in eyes with early to moderate keratoconus. After surgery, the reduction of astigmatism and corneal high-order aberrations allows for an improvement in visual function that is clinically and highly significant.

## Figures and Tables

**Figure 1 jcm-14-00870-f001:**
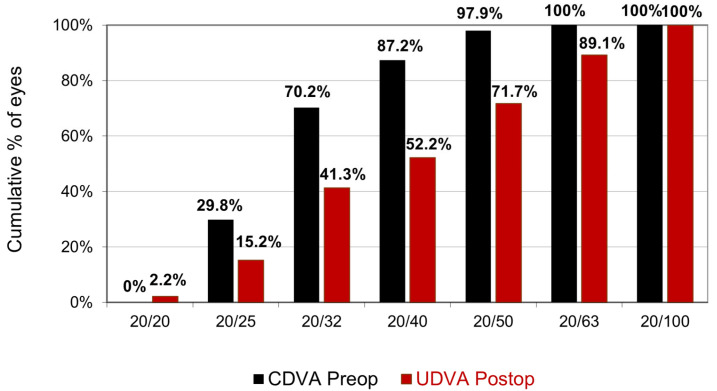
Cumulative percentage of preoperative CDVA versus postoperative UDVA.

**Figure 2 jcm-14-00870-f002:**
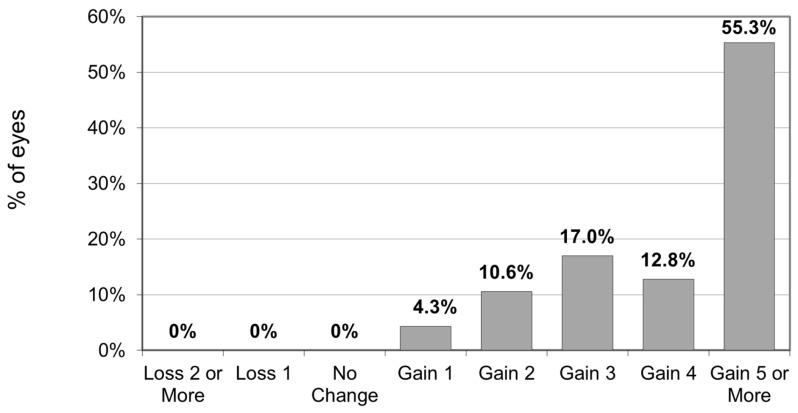
Changes in UDVA Snellen Lines.

**Figure 3 jcm-14-00870-f003:**
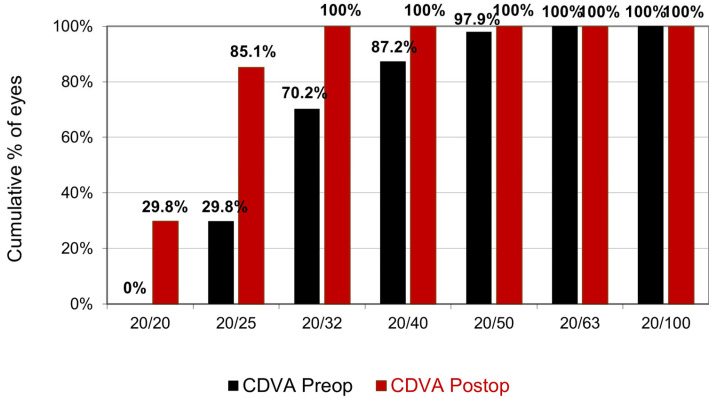
Cumulative percentage of preoperative CDVA versus postoperative CDVA.

**Figure 4 jcm-14-00870-f004:**
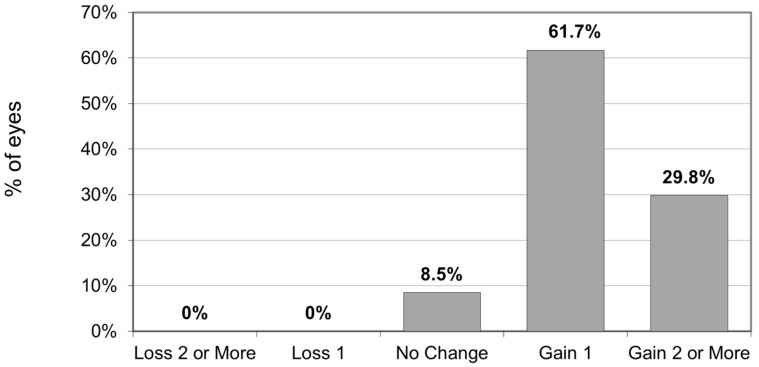
Changes in CDVA Snellen Lines.

**Figure 5 jcm-14-00870-f005:**
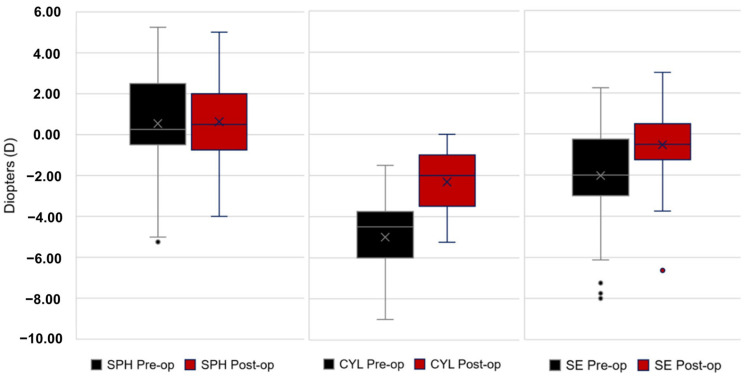
Preoperative and postoperative sphere, cylinder, and sphero-equivalent data.

**Figure 6 jcm-14-00870-f006:**
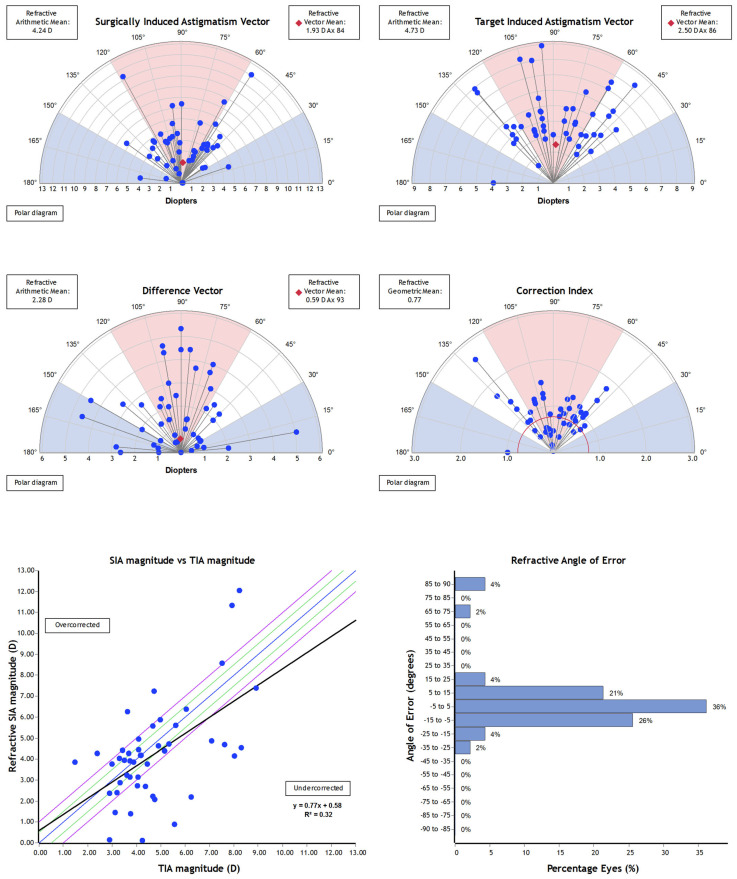
Overview of Alpins vectorial analysis.

**Figure 7 jcm-14-00870-f007:**
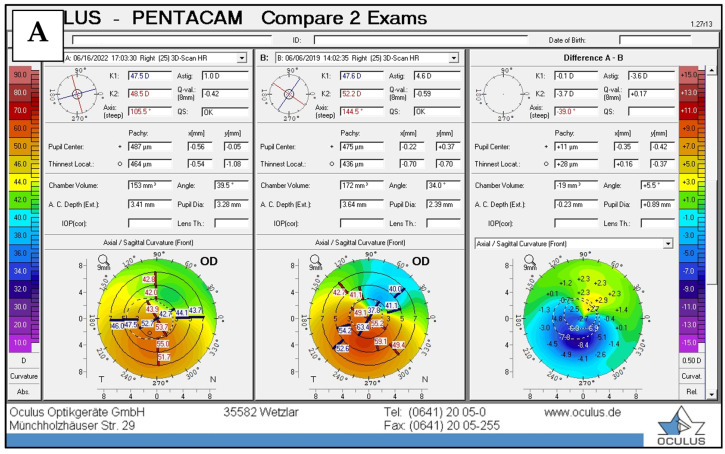
Examples of differences between postoperative vs. preoperative axial curvature maps in eyes with topographic patterns such as a croissant (**A**), duck (**B**), snowman (**C**), and bowtie (**D**).

**Table 1 jcm-14-00870-t001:** Keraring model characteristics.

Topographic Pattern N. of Eyes (%)	Type Symmetric (SI) Asymmetric (AS)	Arc Length (mm)/Thickness (Microns)	
		First ICRS (N. of Eyes)	Second ICRS (N. of Eyes)
Croissant 28 (59.8)	SI	160/300 (11)	90/150 (2), 90/200 (2)
	SI	150/300 (9)	90/150 (1)
	SI	160/250 (5)	
	SI	150/250 (2)	
	SI	150/350 (1)	
Duck 11 (23.4)	AS	160/150–250 (5)	
	AS	160/200–300 (3)	
	AS + SI	160/200–300 (1)	120/250 (1)
	AS	150/150–250 (1)	
	AS	150/200–300 (1)	
Snowman			
Type a 1 (2.1)	AS	160/150–250 (1)	
Type b 6 (12.8)	SI	210/200 (1)	
	SI	210/250 (3)	
	SI	210/300 (2)	
Bowtie 1 (2.1)	SI	120/200 (1)	120/200 (1)

**Table 2 jcm-14-00870-t002:** Keratometric and pachymetric data.

Parameter	Preoperative	Postoperative	Difference	*p*-Value
	Mean ± SD (Min, Max)	Mean ± SD (Min, Max)	Mean ± SD (Min, Max)	
K1 (D)	46.23 ± 2.57 (41.2, 51.4)	44.62 ± 2.17 (41.2, 48.9)	−1.61 ± 1.64 (−7.00, 0.8)	0.001
K2 (D)	49.77 ± 2.53 (43.7, 54.8)	47.17 ± 2.27 (42.2, 52.5)	−2.59 ± 1.78 (−7.1, 0.7)	<0.001
Kmean (D)	48.00 ± 2.37 (42.90, 51.80)	45.89 ± 2.09 (41.90, 50.70)	−2.10 ± 1.42 (−7.05, 0.50)	<0.001
K1—K2 (D)	−3.54 ± 1.92 (−7.40, −0.40)	−2.56 ± 1.51 (−5.80, −0.10)	0.98 ± 1.91 (−2.7, 6.1)	0.007
Kmax (D)	57.63 ± 4.50 (46.50, 68.00)	54.61 ± 3.83 (46.00, 62.20)	−3.02 ± 3.68 (−10.5, 3.6)	<0.001
Thinnest location (μm)	462.8 ± 26.65 (415, 536)	476.4 ± 27.55 (410, 529)	13.7 ± 17.73 (−42, 56)	0.016
Asphericity (Q value at 8 mm)	−0.83 ± 0.32 (−1.44, 0.00)	−0.55 ± 0.33 (−1.36, 0.17)	0.28 ± 0.38 (−1.36, 1.34)	<0.001

SD: standard deviation; K1: corneal power in the flattest meridian; K2: corneal power in the steepest meridian; Kmean: mean corneal power; Kmax: maximum keratometric value.

**Table 3 jcm-14-00870-t003:** Preoperative and postoperative aberrometric data.

Parameter	Preop	Postop	Difference	*p*-Value
	Mean ± SD (Min, Max)	Mean ± SD (Min, Max)	Mean ± SD (Min, Max)	
HOA (μm)	3.296 ± 1.180 (1.161, 7.585)	2.192 ± 0.919 (0.982, 5.050)	−1.103 ± 0.764 (−2.588, 0.339)	<0.001
Coma 90° (μm)	−2.656 ± 1.189 (−6.524, 0.118)	−1.427 ± 1.024 (−4.158, 0.069)	1.228 ± 0.752 (−0.049, 2.669)	<0.001
Coma 180° (μm)	−0.179 ± 1.309 (−2.759, 3.365)	−0.064 ± 0.772 (−1.848, 2.404)	0.115 ± 0.716 (−1.231, 1.646)	0.605
Trefoil 90° (μm)	0.091 ± 0.394 (−1.020, 0.975)	−0.030 ± 0.559 (−0.963, 2.208)	0.135 ± 0.527 (−1.433, 1.464)	0.231
Trefoil 180° (μm)	0.004 ± 0.531 (−0.596, 2.648)	0.007 ± 0.493 (−1.089, 1.521)	−0.005 ± 0.511 (−1.127, 0.881)	0.977
Spherical aberration (μm)	−0.567 ± 0.666 (−1.818, 0.988)	−0.516 ± 0.605 (−1.783, 1.229)	0.039 ± 0.399 (−0.826, 0.946)	0.700

SD: standard deviation; HOA: high order aberrations.

**Table 4 jcm-14-00870-t004:** Preoperative and postoperative topometric indices.

Parameter	Preoperative	Postoperative	Difference	*p*-Value
	Mean ± SD (Min, Max)	Mean ± SD (Min, Max)	Mean ± SD (Min, Max)	
ISV	104.63 ± 28.43 (48, 179)	77.04 ± 25.09 (36, 142)	−26.98 ± 19.46 (−66, 7)	<0.001
IVA (mm)	1.20 ± 0.40 (0.41, 2.22)	0.84 ± 0.38 (0.32, 1.85)	−0.34 ± 0.22 (−0.92, 0.00)	<0.001
KI	1.28 ± 0.11 (1.03, 1.59)	1.17 ± 0.10 (1.01, 1.45)	−0.11 ± 0.06 (−0.30, −0.01)	<0.001
CKI	1.08 ± 0.05 (0.95, 1.17)	1.07 ± 0.04 (0.96, 1.16)	−0.002 ± 0.02 (−0.07, 0.07)	0.849
IHA (μm)	30.08 ± 21.73 (1.4, 81.8)	30.79 ± 19.68 (1.9, 83.5)	0.21 ± 24.62 (−66.2, 44.4)	0.869
IHD (μm)	0.168 ± 0.055 (0.060, 0.321)	0.106 ± 0.046 (0.034, 0.233)	−0.062 ± 0.041 (−0.142, 0.005)	<0.001
Rmin (mm)	5.89 ± 0.47 (4.97, 7.26)	6.20 ± 0.42 (5.42, 7.09)	0.30 ± 0.39 (−0.38, 1.09)	0.001
TKC	2.73 ± 0.55 (1.5, 3.5)	2.13 ± 0.61 (1.0, 3.5)	−0.58 ± 0.53 (−2.0, 0.0)	<0.001

SD: standard deviation; ISV: index of surface variance; IVA: index of vertical asymmetry; KI: keratoconus index; CKI: center keratoconus index; IHA: index of height asymmetry; IHD: index of height decentration; Rmin: radius of minimum axial/sagittal curvature; TKC: topographical keratoconus classification.

## Data Availability

The datasets analysed during the current study are available from the corresponding author on reasonable request.

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
