# Peer review of "Topographic Pattern-Based Nomogram to Guide Keraring Implantation in Eyes with Mild to Moderate Keratoconus: Visual and Refractive Outcome"

_jcm, 2025, doi:10.3390/jcm14030870_

Round 1
Reviewer 1 Report
Comments and Suggestions for Authors
The authors describe outcomes of ICRS insertion using a pattern based nomogram. Generally, the article is very well written, the study is methodical, and the results are clinically relevant. The mean duration of follow-up is good considering the difficulties with follow-up in the keratoconic population.
Methods
- Please indicate whether patients have had a history of cross linking. Did any patients undergo cross linking prior to ICRS insertion, and did any patients undergo crosslinking after ICRS insertion as this may have affected the results?
- Is there any information regarding the history of contact lens use, eg RGP, scleral lenses, soft contacts?
Results
Line 105 - Please indicate how many cases were unilateral and how many cases were bilateral
Consider including the proportion of patients who had greater than 12 months followup.
Was there any correlation between keratoconus severity and the change in CDVA/UDVA postoperatively?
Were there any trends in which type of preoperative tomographic pattern tended to have more improvement in CDVA/UDVA, and which ones had worse outcomes?
Discussion
Succinct and easy to follow
Figures
Consider including one or two figures with examples of pre-op and post-op tomography for various patterns
General comments
some typos/grammatical errors
- line 41 diameter spelled "diametr"
- line 49 there should be a comma after "in this study"
Author Response
The authors describe outcomes of ICRS insertion using a pattern-based nomogram. Generally, the article is very well written, the study is methodical, and the results are clinically relevant. The mean duration of follow-up is good considering the difficulties with follow-up in the keratoconic population.
Methods
Comments 1: Please indicate whether patients have had a history of cross linking. Did any patients undergo cross linking prior to ICRS insertion, and did any patients undergo crosslinking after ICRS insertion as this may have affected the results?
Response 1: Eyes included in the analysis did not receive corneal collagen cross-linking before, in combination or after implantation. It has been reported in the method section (page 2, line 62) and in the results section (page 3, line 112).
Comments 2: Is there any information regarding the history of contact lens use, eg RGP, scleral lenses, soft contacts?
Response 2: We have now specified that all eyes included in the study were contact lens intolerant (page 2; line 61).
Results
Comments 3: Line 105 - Please indicate how many cases were unilateral and how many cases were bilateral.
Response 3: The number of patients operated in one or both eyes has been reported in page 3, line 115.
Comments 4: Consider including the proportion of patients who had greater than 12 months follow-up.
Response 4: The proportion of eyes with follow-up greater than 12 months has been reported in page 3, line 118.
Comments 5: Was there any correlation between keratoconus severity and the change in CDVA/UDVA postoperatively? Were there any trends in which type of preoperative tomographic pattern tended to have more improvement in CDVA/UDVA, and which ones had worse outcomes?
Response 5: The correlation between keratoconus severity/topographic pattern and postoperative visual outcome was not investigated because we thought that the small sample size did not allow to address this issue. This is a limitation of the study that has been added in page 11, line 349.
Discussion
Succinct and easy to follow.
Figures
Comments 6: Consider including one or two figures with examples of pre-op and post-op tomography for various patterns.
Response 6: We have added the Figure 7 that shows post-operative tomographic changes in eyes with different preoperative topographic pattern.
General comments
some typos/grammatical errors
Comments 7: - line 41 diameter spelled "diametr"; - line 49 there should be a comma after "in this study".
Response 7: We have made the requested corrections.
Reviewer 2 Report
Comments and Suggestions for Authors
Congratulations to the authors on this well-written paper. However, the use of Kerarings has been extensively documented in the literature for over 15 years. The manuscript lacks novelty.
Following you will find my answers to the questions: 1. What is the main question addressed by the research? The assessment of outcomes of Kerning implantation based on topographic patterns. 2. What parts do you consider original or relevant to the field? What specific gap in the field does the paper address? Keraring implantation is a well-documented procedure in the literature, with consistently positive outcomes. A longer follow-up time after Keraring implantation could be adressed from this paper. 3. What does it add to the subject area compared with other published material? To my opinion a longer follow-up time compared to other published material, for example already published material: 10.1055/a-0659-2549: smaller number of patients, shorter follow up 10.1007/s40123-017-0117-3 smaller number of patients, shorter follow up 10.2147/OPTH.S120267smaller number of patients, shorter follow up 10.2147/OPTH.S333832smaller number of patients, shorter follow up , combined with CXL 10.1155/2017/4313784 larger number of patients, shorter follow-up And the list goes on. 4. Are the conclusions consistent with the evidence and arguments presented? Yes Were all the main questions posed addressed? By which specific experiments? Yes 5. Are the references appropriate? Yes
6. Any additional comments on the tables and figures and the quality of the data.
Author Response
Congratulations to the authors on this well-written paper. However, the use of Kerarings has been extensively documented in the literature for over 15 years. The manuscript lacks novelty.
Following you will find my answers to the questions:
Comments 1: What is the main question addressed by the research? The assessment of outcomes of Kerning implantation based on topographic patterns.
Comments 2: What parts do you consider original or relevant to the field? What specific gap in the field does the paper address? Keraring implantation is a well-documented procedure in the literature, with consistently positive outcomes. A longer follow-up time after Keraring implantation could be adressed from this paper.
Comments 3: What does it add to the subject area compared with other published material? To my opinion a longer follow-up time compared to other published material, for example already published material:
- 1055/a-0659-2549 smaller number of patients, shorter follow up
- 1007/s40123-017-0117-3 smaller number of patients, shorter follow up
- 2147/OPTH.S120267 smaller number of patients, shorter follow up
- 2147/OPTH.S333832 smaller number of patients, shorter follow up, combined with CXL
- 1155/2017/4313784 larger number of patients, shorter follow-up
And the list goes on.
Response 1, 2, 3: We fully agree with the reviewer that Keraring implantation for keratoconus treatment has been already documented in the literature. The manuscript assesses the outcome of the procedure using a nomogram that the manufacturer has recommended to use since 2018. To the best of our knowledge, only 2 studies had reported the efficacy and safety of the procedure using that nomogram. Those studies included keratoconic eyes with asymmetrical topographic pattern. The cohort of eyes analyzed in our study also included eyes with symmetrical pattern. Including both symmetrical and asymmetrical patterns allows to estimate the overall performance of this method in eyes with mild to moderate keratoconus (page 2, line 52).
Are the conclusions consistent with the evidence and arguments presented? Yes
Were all the main questions posed addressed? By which specific experiments? Yes
Are the references appropriate? Yes
Any additional comments on the tables and figures and the quality of the data.
Reviewer 3 Report
Comments and Suggestions for Authors
an interesting topic considering the proportion of pac with keratoconus. At the introduction if possible more references and more information about the most recently used intracorneal rings after a more detailed history of them.
Material and method, the inclusion and exclusion criteria are clear. Possibly if there is a scheme that is easier to follow. The results are as expected. Even if there is a small number of patients, the results are statistically significant
The conclusions must be reformulated. Tb to reflect the results. If the authors can highlight the importance of these rings in the treatment of early keratoconus and future directions according to the results
kind regard
Author Response
An interesting topic considering the proportion of pac with keratoconus.
Comments 1: At the introduction if possible more references and more information about the most recently used intracorneal rings after a more detailed history of them.
Response 1: The most used intracorneal rings internationally available have been reported in page 1, line 38.
Comments 2: Material and method, the inclusion and exclusion criteria are clear. Possibly if there is a scheme that is easier to follow.
Response 2: We tried to provide a detailed description of the preoperative protocol and surgical technique. However, we avoided to duplicate the full nomogram guidelines which can be consulted on-line as reported in the reference n.12.
The results are as expected. Even if there is a small number of patients, the results are statistically significant.
Comments 3: The conclusions must be reformulated. Tb to reflect the results. If the authors can highlight the importance of these rings in the treatment of early keratoconus and future directions according to the results
Response 3: The conclusion has been reformulated to reflect the results of the study (page 12, line 354).
Round 2
Reviewer 2 Report
Comments and Suggestions for Authors
Congratulations to the authors for the changes.